# Root and Leaf Anatomy, Ion Accumulation, and Transcriptome Pattern under Salt Stress Conditions in Contrasting Genotypes of *Sorghum bicolor*

**DOI:** 10.3390/plants12132400

**Published:** 2023-06-21

**Authors:** Appa Rao Karumanchi, Pramod Sivan, Divya Kummari, G. Rajasheker, S. Anil Kumar, Palakolanu Sudhakar Reddy, Prashanth Suravajhala, Sudhakar Podha, P. B. Kavi Kishor

**Affiliations:** 1Department of Biotechnology, Acharya Nagarjuna University, Nagarjuna Nagar, Guntur 522 209, India; raokarumanchi66@gmail.com; 2Department of Chemistry, Division of Glycoscience, KTH Royal Institute of Technology, School of Engineering Sciences in Chemistry, Biotechnology and Health, Albanova University Center, SE-10691 Stockholm, Sweden; psivan@kth.se; 3International Crops Research Institute for the Semi-Arid Tropics (ICRISAT), Patancheru, Hyderabad 502 324, India; divyanu.k@gmail.com (D.K.); sudhakarreddy.palakolanu@icrisat.org (P.S.R.); 4Department of Genetics, Osmania University, Hyderabad 500 007, India; rajguddimalli@yahoo.in; 5Department of Biotechnology, Vignan’s Foundation for Science, Technology & Research (Deemed to Be University), Guntur 522 213, India; anilkumarou@gmail.com; 6Amrita School of Biotechnology, Amrita Vishwa Vidyapeetham, Kollam 690 525, India; prash@am.amrita.edu

**Keywords:** ion accumulation, root and leaf anatomy, sorghum, salt stress, transcriptome

## Abstract

Roots from salt-susceptible ICSR-56 (SS) sorghum plants display metaxylem elements with thin cell walls and large diameter. On the other hand, roots with thick, lignified cell walls in the hypodermis and endodermis were noticed in salt-tolerant CSV-15 (ST) sorghum plants. The secondary wall thickness and number of lignified cells in the hypodermis have increased with the treatment of sodium chloride stress to the plants (STN). Lignin distribution in the secondary cell wall of sclerenchymatous cells beneath the lower epidermis was higher in ST leaves compared to the SS genotype. Casparian thickenings with homogenous lignin distribution were observed in STN roots, but inhomogeneous distribution was evident in SS seedlings treated with sodium chloride (SSN). Higher accumulation of K^+^ and lower Na^+^ levels were noticed in ST compared to the SS genotype. To identify the differentially expressed genes among SS and ST genotypes, transcriptomic analysis was carried out. Both the genotypes were exposed to 200 mM sodium chloride stress for 24 h and used for analysis. We obtained 70 and 162 differentially expressed genes (DEGs) exclusive to SS and SSN and 112 and 26 DEGs exclusive to ST and STN, respectively. Kyoto Encyclopaedia of Genes and Genomes (KEGG) and Gene Ontology (GO) enrichment analysis unlocked the changes in metabolic pathways in response to salt stress. qRT-PCR was performed to validate 20 DEGs in each SSN and STN sample, which confirms the transcriptomic results. These results surmise that anatomical changes and higher K^+^/Na^+^ ratios are essential for mitigating salt stress in sorghum apart from the genes that are differentially up- and downregulated in contrasting genotypes.

## 1. Introduction

Salinity is one of the major abiotic stresses that result in reduced plant growth and productivity through diminishing water potential [1,2]. Around 1000 million hectares of land is being rendered unsuitable for cultivation by salinity every year [3]. To make crop plants grow with superior performance under these perturbed conditions, we need to unravel first the underlying physiological and molecular mechanisms associated with salt stress tolerance [4]. Salt stress enhances the build-up of reactive oxygen species (ROS) [5] in addition to osmolytes [6]. Salinity induces membrane damage, enzyme and photosynthesis inhibition, nutrient unavailability, and altered levels of growth regulators, resulting in the death of the plants [7]. Anatomical adaptations to imposed salt stress have been noticed in many plants. Notable among them were reduced leaf thickness, enhanced cell vacuolar volume, aerenchyma formation, and high sclerification. Such features have been recorded in *Typha domingensis* under salt stress conditions [8]. In wheat, reduced stem and leaf diameter, wall thickness, diameter of the hollow pith cavity, and increased phloem thickness and diameter of the metaxylem vessel were recorded [9]. Barley (*Hordeum brevisubulatum*) plants exposed to salt stress display decreased vascular tissues in leaf, stem, and root, aside from diminished thickness of epidermis in stems and leaves. Imposition of sodium chloride stress enhances the thickness of the endodermis and epidermis, indicating that anatomical changes occur under salinity stress [10]. In salt-sensitive *Populus canescens*, moderate salt stress increases the vessel frequencies but decreases vessel lumina. In *P. euphratica*, a salt-tolerant genotype, a set of genes was suppressed under salt stress conditions, but without any anatomical changes. Exposure to salt impacts the composition of cell walls, including an increase in lignin: carbohydrate ratio in both species [11]. In general, anatomical changes appear to be crucial to reduce evapotranspiration and restrict Na^+^ entry into leaf cells under saline conditions.

Salinity causes ionic stresses brought about by the accumulation of excess Na^+^ ions in the stems and leaves [12]. Excess Na^+^ ions in the cytoplasm cause ionic imbalance and lead to perturbation in the Na^+^/K^+^ ratio. Proper maintenance of Na^+^ and K^+^ levels under sodium chloride stress is crucial for salt stress tolerance [13]. Plants maintain a high K^+^/Na^+^ ratio for cellular maintenance and electroneutrality, but this ratio is disturbed by high concentrations of Na^+^ and Cl^−^ ions in soil and also due to leakage of K^+^ from root cells [7,14,15]. For achieving ionic homeostasis in cells, or low Na^+^ accumulation in leaves, proper control between Na^+^ and K^+^ transporters and/or channels is imperative [16,17,18,19,20,21,22,23]. Stable plasma membrane potentials also appear to be highly necessary for proper ion homeostasis alongside the activity of the salt overly sensitive (SOS1) pathway, haem oxygenase involvement, and regulation of SOS1 expression [12,23,24]. Besides Na^+^ and K^+^, calcium (Ca^2+^) levels and signalling are also altered, and such changes are associated with salt stress tolerance [21].

A reprogramming of the plant gene expressions under saline conditions is crucial for modulating a multitude of anatomical, physiological, and biochemical processes. Reprogramming of the gene expressions might fluctuate in diverse genotypes to confer salt stress tolerance. Plants combat salt stress through signal transduction and response through modulating biochemical, physiological, and molecular activities [4,25,26]. Salt stress causes massive changes in gene expressions; for example, in sugar beet, 58 unigenes which include 14 singletons and 44 contigs were noticed as differentially expressed genes. Salt-responsive genes that are involved in energy, metabolism, photosynthesis, protein degradation and synthesis, stress, and defence were also noticed [27]. Transcriptomic profiles were studied after exposing the seedlings of watermelon to 300 mM sodium chloride for a short term [28]. Analysis of the transcriptome revealed 7662 DEGs, out of which 4055 were found to be upregulated. Further, they noticed 240 upregulated and 194 downregulated differentially expressed transcription factors. Among them, ethylene response factors (ERFs), WRKY, NAM, ATAF and CUC (NAC), basic helix-loop helix (bHLH), and myeloblastosis viral oncogene homolog (MYB)-related families were recorded [28].

*Sorghum bicolor* (L.) Moench is the world’s fifth most important cereal crop and is cultivated in the arid and semi-arid regions of South Africa, Australia, India, and America [29]. It is mostly grown for its grain and fodder, and the crop serves as a staple food for 500 million people. Salt stress inhibits germination and also seedling growth in *S. bicolor* [30]. Quantitative trait loci (QTLs) associated with salt stress tolerance at the germination and seedling stage have been identified [31]. Genotypic differences in sorghum also exist for the tolerance of salt stress. Information on the physiology of salt stress tolerance in *S. bicolor* is known [32,33]. However, it is not known if anatomical changes exist in the roots and leaves of the sorghum genotypes that differ in their salt sensitivity levels. Further, cation and anion accumulation patterns and transcriptomic changes, if any, have not been thoroughly investigated between the salt-susceptible and salt-tolerant lines of grain sorghum under short-term sodium chloride exposure. Accordingly, the leaves and roots of salt-susceptible ICSR-56 (SS), salt-susceptible genotype treated with 200 mM sodium chloride (SSN), salt-tolerant CSV-15 (ST), and salt-tolerant genotype treated with sodium chloride (STN) have been used for finding out the differences in the leaf and root anatomy, ion accumulation patterns, and differentially expressed genes among the two lines. Transcriptomic profiling in relation to heat and drought stress was studied in grain sorghum [34] and in two sweet sorghum genotypes with different salt tolerance abilities [35,36]. Therefore, in the present study, we examined the changes in gene expressions during short-term salt stress exposure of sorghum seedlings using RNA sequencing (RNA-seq) and identified candidate genes and pathways. We noticed structural differences in the root and leaf anatomy and significant differences in ion accumulation patterns in different organs of the contrasting genotypes of sorghum under salt stress conditions, which might help breeders to use such traits in their breeding programs.

## 2. Results

### 2.1. Leaf Anatomy in SS and ST Lines under Control and Stress Conditions

Leaf anatomy has been compared in the present study between salt-susceptible (SS) leaf control (abbreviated as SSL-C), SS leaf treated with 200 mM sodium chloride (SSL-T), salt-tolerant (ST) leaf control (STL-C), and ST leaf treated with 200 mM sodium chloride (STL-T). SS plants showed a relatively thinner cell wall on the upper side of the epidermis (Figure 1B), suggesting a variation in the thickness of the cuticle layer compared to that of ST plants (Figure 1H). The vascular bundles in the mid rib showed metaxylem vessels with a large lumen diameter and thin cell walls (Figure 1A,C) compared to thick-walled elements in the ST plants (Figure 1I). Salinity stress causes accumulation of polysaccharides in the cell corners of phloem elements within the vascular bundles of SS plants (Figure 1F). The lignin distribution in the secondary cell wall of sclerenchymatous cells beneath the lower epidermis has been found to be significantly higher in the ST leaf tissue after 200 mM sodium chloride treatment (Figure 1L). The cuticle layer also showed an increased thickness in the upper epidermis of ST leaves under 200 mM sodium chloride stress (Figure 1K) when compared with SS leaves (Figure 1E).

### 2.2. Root Anatomy in SS and ST Lines under Stress Conditions

Root anatomy has been compared between SS root control (abbreviated as SSR-C), SS root treated with 200 mM sodium chloride (SSR-T), ST root control (STR-C), and ST root treated with 200 mM sodium chloride (STR-T). In comparison with the SS roots (Figure 2A,B), roots of ST plants showed an increase in the number of cell layers with thick, lignified walls constituting the hypodermis (Figure 2G,H). The characteristic wall thickening on the endodermis was also higher in ST roots (Figure 2I). The SS plants were characterized by metaxylem elements with thin cell walls and a large lumen diameter (Figure 2C) compared to relatively thick-walled narrow-lumen xylem elements in ST roots (Figure 2I). Cuticles are thin in SS plants in comparison with ST plants. After salt treatment, STN plants displayed an increase in lignified cell walls of the hypodermis (Figure 2J,H) while SSN plants did not exhibit any variation in cell wall structure of the hypodermis (Figure 2D,E). The cell corner region of cortical cells often showed accumulation of polysaccharides in STN (Figure 2D). In both SSN and STN plants, salinity stress exhibited an increase in the thickness of lower tangential and radial walls with Casparian strips in the endodermis (Figure 2F,L). The ST roots were distinguished by a greater amount of lignin in the thickened wall areas. The SS root, on the other hand, was characterized by the occurrence of tylosis in the metaxylem elements (Figure 2F). The large lumen metaxylem elements with thin cell walls often deformed and collapsed following salt treatment in the SS genotype (Figure 2D). Vascular bundles exhibit a large lumen diameter with thin cell walls in SS, compared to thick-walled elements in ST.

### 2.3. Ultra-Structural Changes in the Leaf Cuticle

The transmission electron micrographic (TEM) observations of the ST leaf revealed the occurrence of a relatively thick cuticle on the upper epidermis (Figure 3A) compared to that of the SS leaf (Figure 3B). The cuticle thickness was found to be higher in the STN leaf in contrast to thinner cuticle in the SSN leaf, suggesting that the cuticle has a significant role to play in salinity stress tolerance in sorghum.

### 2.4. Ultra-Structural Changes and Lignin Distribution Pattern in the Root

The transmission electron microscopy (TEM) analysis of root sections revealed the ultra-structural changes in the cell wall and lignin distribution pattern in SS and ST plants. Roots of ST plants unveiled an increase in Casparian thickening (Figure 4A) in comparison with that of SS roots (Figure 4D). Salt treatment resulted in an enhancement of thickening to the tangential wall of endodermal cells of STN roots (Figure 4B). Presence of lignin and its homogenous distribution within the thickening was evident from the electron dense regions found in the Casparian thickening of STN roots. The SSN root shows enhanced thickening in an abnormal pattern (Figure 4E), with lignin distribution as different layers within the thickening (Figure 4F) indicating an inhomogeneous distribution and chemical constitution of Casparian thickening in the root. The metaxylem vessel element in the STN root showed a thick, multi-layered secondary wall having a more or less homogeneous distribution of lignin (Figure 5A). A lignin-rich layer was evident in the boundary between two sublayers found within the S2 layer of secondary vessel walls (Figure 5A). In comparison, the cell wall of the metaxylem vessel element in the SSN root was thinner and revealed a high inhomogeneity in lignin distribution within the secondary wall. The S1 and S2 layers unveiled several pockets of less lignin distribution (Figure 5C), suggesting weak cell wall integrity in the deformed vessels due to the weak/abnormal lignification pattern in SSN plants.

### 2.5. Accumulation of Ions in Different Organs in SS, SSN, ST and STN Plants

Ion accumulation in different tissues of plants exposed to salt stress is a universal phenomenon. This is a consequence of perturbations in the cell wall, which modulates downstream genes that help in the accumulation of cations such as sodium (Na^+^). Overall, it appears that salt-susceptible lines accumulate higher levels of Na^+^ when compared to potassium (K^+^) (Figure 6). In *S. bicolor*, SS accumulated higher Na^+^ (Figure 6A) and lower levels of K^+^ (Figure 6B) when compared to ST, irrespective of the tissue type, such as root, stem, and leaf, and treatment (control versus salt stress). Such a marked difference in the tissues and genotypes was not observed for calcium (Ca^2+^) (Figure 6C) and chloride (Cl^−^) levels (Figure 6D).

### 2.6. Identification of DEGs, KEGG, and GO Pathways

Paired-end reads used for reference-based read mapping are shown in Appendix A. The transcriptomic data obtained for SS, SSN, ST, and STN treatments are represented in Appendix A, respectively. Significant DEGs associated with SS/SSN (Figure 7A) and ST/STN (F Figure 7B) pairs were identified across the inferred datasets. Seventy genes are exclusive to SS: 162 for SSN, in contrast to 112 and 26 for ST and STN, respectively. Likewise, 668 and 250 have been found to be upregulated, while 310 and 769 are downregulated in SS and ST genotypes (Figure 7A,B). In SS/SSN genotypes, 21,473 genes were expressed, and 24,405 in ST/STN (Figure 7A,B). The KEGG revealed diverse pathway genes associated with carbohydrate, energy, lipid, nucleotide, amino acid, metabolism of cofactors and vitamins, terpenoids, polyketides, and secondary metabolism in SS, SSN, ST, and STN samples (Figure 8). Likewise, genes involved in gene information processing; environmental information processing, including signal transduction’ cellular processes’ and organismal systems have been detected (Figure 8). GO analysis disclosed the genes implicated in biological processes, cellular components, and molecular function (Figure 8). The striking difference is that a higher number of genes have been found downregulated in the ST genotype in comparison with SS. The difference is pronounced in all the areas including biological processes (83 vs. 299), cellular components (81 vs. 321), and molecular function (118 vs. 343) in SS vis à vis ST genotypes (Table 1).

### 2.7. DEGs in SS and ST Genotypes under Control and Salt-Treatment

In salt-susceptible genotypes and salt-tolerant genotypes (without sodium chloride treatment), 70 and 112 DEGs were recorded, respectively. Between the control and tolerant and susceptible pairs, a large number of DEGs were associated and the common among them was found to be glucan endo-1,3-beta-glucosidase 3-like genes and a few uncharacterized proteins (Appendix A). Interestingly, among the top DEGs, one miRNA was seen upregulated, indicating that genetic variation is best seen with miRNAs as well. Few long non-coding RNAs (lncRNAS) are anticipated beyond the intended cut off. While endo-1,3-beta-glucosidase 3-like genes are known to be pathogenic in nature, they are an important class of hydrolytic enzymes that are abundant in many plants associated with stress. They are also known to be associated both with defence reactions and abiotic stresses along with chitinases after infection. Glucan endo-1,3-beta-glucosidase 3-like, BARWIN, peroxidases, chitinase 2, receptor kinase-like protein XA 21, and heavy metal-associated isoprenylated plant protein 16 (named as HIPP16) have been included among the top DEGs in SS/SSN for further validation using qRT-PCR. The receptor kinases, chitinase, and peroxidase are also associated with susceptible genotype SS, but what emerged perhaps as an interesting candidate is the class of metalloprotein HIPPHIPPs, which contains two domains: one metal binding domain at the N terminus of the protein and an isoprenylated motif at the C terminal. Over the years, their role in heavy metal homeostasis and stress susceptibility has largely been deciphered. It is inferred that this class of proteins is noticed in vascular plants and during environmental changes. The HIPP gene is downregulated in both SS/SSN and ST/STN samples, and this begs the question of whether metal-susceptible stress factors are associated with them, which invariably is unique to a set of DEGs between the two samples. From the 27 top DEGs in SS/SSN that are chosen, only three genes have been found in S. bicolor from our ID mapping (Appendix A), wherein tubulin-regulated genes are notably enriched. On the other hand, in ST/STN, six genes have been mapped (Appendix A) such as filament-like plant protein, UDP-glucuronate:xylan alpha-glucuronosyltransferase, nucleotide pyrophosphatase/phosphodiesterase, and CLIP-associated protein in addition to microtubule-associated proteins and beta-glucosidases. However, interactions have not been associated with these proteins as they remain unique and distinct besides a number of uncharacterized sequences between them.

### 2.8. Validation of Selected DEGs by qRT-pCR

In order to functionally utilize the transcriptome data obtained, selected gene expressions (20 genes for each genotype) were validated via qRT-PCR analysis for both SS and ST genotypes, respectively (Table 2). While SS values were used as controls against SSN, ST values were taken as controls for deriving STN values. Expression studies of all the 20 genes (Table 2) were carried out in SS and ST sorghum leaves subjected to salt treatment for 24 h and shown as a heat map (Figure 9 and Figure 10). Differential expression profiles of genes (Table 2) have been observed in SS and ST under sodium chloride treatment. All the genes (Table 2) displayed higher expression levels in ST under sodium chloride treatment (Figure 11). Similarly, all the genes (Table 2) displayed higher expression levels in SS under sodium chloride treatment except Lipid transfer protein (LTP), non-coding RNA (miRf11471-akr), and Fasiclin-like arabinogalactan protein 12 genes, which are downregulated (Figure 11).

## 3. Discussion

### 3.1. Anatomical Modifications in Tolerant and Susceptible Sorghum Genotypes

Plant cuticle is a lipid-rich layer covering the epidermal cell wall of leaves, and it plays a critical role in plant tolerance through its ability to postpone the onset of cellular dehydration during osmotic stress [59,60]. The present study shows an increase in cuticle thickness of salt-tolerant sorghum plants, which is more likely an effective adaptive mechanism to tolerate dehydration under salinity stress compared to that of salt-susceptible plants. Enhancement of hydrophobicity of the cell wall via lignification forms an important strategy to control apoplastic movement of toxic ions into the cell. In *Sorghum*, the salt-tolerant roots showed significantly enhanced lignification in the cell walls of the hypodermis, endodermis, and pericycle. Salt stress is reported to activate lignin deposition and reduce apoplastic transport in soybean roots, which can strengthen the Casparian strip and also act as an alternative hydrophobic barrier to bypass flow [61]. Both secondary cell wall deposition and lignification have affected abiotic stresses such as salt stress [62]. Liu et al. [63] have shown that abscisic acid (ABA) modulates secondary cell wall formation besides the deposition of lignin through phosphorylation of a NAC SECONDARY WALL THICKENING FACTOR 1 (NST1). NST1 is a member of the NAC transcription factor family that manages the transcriptional activation of a suite of downstream genes that are mostly enmeshed in the biosynthesis of cellulose and lignin during stress in dicots [63].

The salt-susceptible roots, although exhibiting slight cell wall thickening in endodermis and pericycle, exhibited less lignin distribution, suggesting the improper incorporation of polysaccharides into the secondary cell wall composition. Salt-tolerant roots in the present study display relatively narrow lumen vessels and thick cell walls that resist the salinity stress associated with xylem pressure, while thin-walled and wide-lumen metaxylem elements in susceptible roots collapse under salinity stress. It appears that salinity induces discontinuous protoxylem via a DELLA-dependent channel to promote salt stress tolerance in Arabidopsis [64]. In poplar, salt stress resulted in enhanced conduit reinforcement, which is more likely an increased safety mechanism against implosion [65]. Cell wall integrity (CWI) is the key for plant adaptation to salt stress. During salt stress, several genes associated with cell wall biosynthesis either up- or downregulate, and their regulation is crucial in cell wall sensing, reduced plant growth or defects, if any, and also overall plant survival [66,67]. In these circumstances, surveillance of the cell wall status and tracking wall-sensing and signalling pathways involved in CWI and wall remodelling is vital for the perception of stress responses for any plant [68,69,70]. It is possible that defective cell wall synthesis in plants leads to salt susceptibility. Importantly, mutations in cell wall sensor protein FERONIA led to cell bursting under salt stress; thus, a defective gene has been linked to defects in wall surveillance and failure to repair damage [71]. Therefore, CWI is vital for plant survival under saline conditions.

The prominent endodermis and pericycle layer in salt-tolerant species of *Lentis* provides an edge for entry for ions compared to salt-sensitive species [72]. Interestingly, salt-tolerant roots in the present study also showed an increase in the lignification pattern in the endodermal cell wall with Casparian strips in response to salinity stress. Tu et al. [73] found that salt-tolerant rice contained less aerenchyma and possessed a thicker barrier in the endodermis. Such a thicker endodermis in tolerant lines blocks the transport of Na^+^ ions to pericycle cells. Thicker cuticles and endodermis have also been noticed in the present study, reasoning that thicker endodermis plays a key role in the restriction of Na^+^ entry into the xylem. Taken together, the anatomical characteristics suggest that the hydraulic conductivity in salt-susceptible *S. bicolor* plants have been greatly compromised under salinity stress due to less lignification, causing a weak CWI.

### 3.2. Estimation of Na^+^ and K^+^ Ions in SS and ST Lines

Salt stress causes ionic toxicity and nutritional disorders, besides disturbing K^+^/Na^+^ ratios. However, plants have evolved strategies such as Na^+^ exclusion, sequestration into vacuoles, retention of K^+^ ions in the cytosol, and restriction of Na^+^ ion loading into the xylem to minimize these perturbations and to survive under salt stress conditions [74,75,76]. Evidence exists that salt-tolerant plants accumulate less Na^+^ in the leaves and shoots in comparison with salt-susceptible plants [77,78]. Salt overly sensitive pathway 1 (SOS1) not only mediates Na^+^ exclusion at the plasma membrane, but also xylem Na^+^ loading [21,79,80]. In salt-tolerant plants, besides low levels of Na^+^, high accumulation of K^+^ has been noticed [81]. K^+^ leaks out through membrane-depolarization-activated outward-rectifying K^+^-selective (GORK) channels [82,83,84,85]. Almodares et al. [86] previously noticed a similar response under salt stress in sweet sorghum lines. Salt tolerance and allocation of Na^+^ ions under soil salinity in sorghum has been investigated by Calone et al. [87]. High soil salinity coupled with water salinity resulted in higher Na^+^ bioaccumulation in stems and leaves, the most sensitive organs. However, salt-tolerant plants, by some means, accumulate more K^+^, or the ratio of K^+^/Na^+^ is always higher than the susceptible lines. In the present study, ST and STN plants accumulated higher levels of K^+^ and lower levels of Na^+^ in root, stem, and leaf organs in comparison with SS and SSN plants (Table 1). The above results, including ours, infer that a higher K^+^/Na^+^ ratio is a critical component for obtaining salt-tolerant plants and also restriction of Na^+^ entry into the most sensitive organs such as leaves. Especially in the ST line, leaf Na^+^ was less in comparison with the SS line, indicating that root-specific anatomy might have restricted Na^+^ entry into leaves. Liu et al. [44] noticed retention of K^+^ in rice root zones, which was conferred by higher H^+^-ATPase, decreased K^+^ efflux, and reduced upregulation of OsGORK along with higher upregulation of OsAKT1 in salt-tolerant lines. Overall, it appears that organ-specific regulation of not only Na^+^ but also K^+^ ion transporters is very vital to explain salt susceptibility versus tolerance in sorghum genotypes.

### 3.3. Transcript Profiling

The availability of this crop’s genome sequence and contrasting genotypes make it possible to elucidate its mechanisms of salt stress tolerance [88]. Salt stress tolerance is controlled by a multitude of genes [89]. Therefore, transcriptomic analysis is an effective approach to distil the genes that are expressed globally under salt stress conditions. Transcriptomic analysis in relation to water use efficiency [90], identification of beta-alanine betaine (an osmolyte involved in abiotic stress) biosynthesis [91], drought stress [92,93], and combined transcriptomic and metabolomics analyses [94] have been performed in sorghum. In the present study, gene ontology (GO) classification has helped us to understand the DEGs in response to salt stress in contrasting genotypes. Several genes involved in carbohydrate, fat, nitrogen, and amino acid metabolism; cellulose synthesis; lignin biosynthesis; hormone and calcium signalling pathways; ion transport; transcription factors; and secondary plant product metabolism have been found enriched in response to salt stress. Many genes such as endo-1-3-beta-glucosidases and chitinases are pathogenesis-related proteins involved in antifungal activities but are also associated with salt stress conditions in barley [95,96]. Genes involved in wall building, such as cellulose synthase and the lignin biosynthetic pathway, have been found modulated under salt stress conditions, indicating that cell wall thickening by means of lignin deposition is crucial for salt stress tolerance [97]. Protein kinases are involved in signal transduction during salt stress and trigger several downstream genes involved in salt stress tolerance [98]. Evidence exists that by some means, lipid transfer proteins (LTPs) are involved in salt stress tolerance [99,100,101]. Our results are consistent with previous studies carried out by Wang et al. [102] in salt-sensitive and salt-tolerant maize and in sorghum exposed to drought stress by Abdel-Ghany et al. [92] and Azzouz-Olden et al. [93]. Salt stress has been found to elicit a wide spectrum of transcriptional changes with an overlap of DEGs. Such a comparison of the DEGs in contrasting genotypes mirrors the mechanism of plant salt tolerance. The results also indicate that major genes and many pathways implicated in salt stress response are conserved across plant species [94,102,103].

### 3.4. Validation of Genes via qRT-PCR

Among the twenty chosen genes for qRT-PCR analysis (20 for SSN and STN each), high transcript levels were noticed for seventeen of them, importantly for BARWIN and peroxidase BARWIN, a member of the pathogenesis-related protein (PR-4) induced in barley via wounding or pathogens and which has similarity with proteins that are encoded by the wound-induced genes of barley (win) and similar other proteins. It is involved in a common defence mechanism in plants [39], and its homologs have been identified in other cereals such as wheat [104], maize [105], and rice [106]. Glucan endo-1,3-beta glucosidase is upregulated in both SSN and STN samples. They play pivotal roles in cell division, trafficking of materials in plasmodesmata, fighting against fungal pathogens along with chitinases, and in abiotic stresses [37,38]. Peroxidase is implicated in antioxidative defence and converts the toxic hydrogen peroxide into water and oxygen. Its overexpression leads to salt stress tolerance in plants [40,107]. Glucan endo-1,3-beta glucosidase and peroxidases are upregulated in both the genotypes, indicating the key roles they play in salt stress response. Other important proteins that are upregulated in SSN samples are the K^+^ transporter and pathogenesis/defence-related ones. K^+^ uptake during salt stress is crucial for maintaining ion homeostasis [23]. In STN, the pyrophosphatase gene showed high transcript levels. Vacuolar H^+^ pyrophosphatase (H^+^-PPase) catalyses pyrophosphate into phosphate, which is coupled with proton translocation across the vacuolar membrane. H^+^-PPase gene overexpression has been shown to improve plant growth through enhancing cell number [108] and to alleviate salt and drought stresses in finger millet, sugarcane, and Arabidopsis [109,110,111]. HHO5 has been shown to regulate flowering under stress conditions in addition to mitigating stress [112]. While the ncRNA is downregulated in SSN, it is upregulated in STN, inferring that it is associated with salt stress in the tolerant genotype. These studies indicate that several of the genes that are upregulated under salt stress conditions work in synergy in the network and modulate their responses.

It is concluded that significant changes in the leaf and root anatomy occur besides ion accumulation patterns and transcriptomic changes in contrasting genotypes of grain sorghum. Such traits can be used in breeding programs aimed at developing salt-tolerant genotypes.

## 4. Material and Methods

### 4.1. Sample Preparation for Microscopy

For taking anatomical sections, to study ion accumulation patterns and tracriptional changes, salt-susceptible (SS) and salt-tolerant (ST) sorghum genotypes ICSR-56 and CSV-15, respectively were used. Suitably trimmed pieces of leaf, root, and stem pieces were fixed in a mixture of 0.1% glutaraldehyde and 4% paraformaldehyde in 50 mM sodium cacodylate buffer for 4 h at room temperature. After washing in the buffer, tissues were dehydrated in graded series of ethanol (30–95%, 15 min each, pure ethanol × 3, each for 20 min) and embedded in London Resin (LR) white, as described elsewhere [113]. For light microscopy, semi-thin sections (1–2 µm) were taken from LR white embedded blocks with a glass knife using Leica ultramicrotome (Leica UMO7, Germany). Sections were stained with 0.05% aqueous toluidine blue O [114] and mounted in DPX. Stained sections were observed and photographed using Leica DM200 microscope (Germany). For transmission electron microscopy (TEM), ultrathin sections of 90 nm thickness were taken from the LR white embedded blocks and mounted on nickel grids. For leaf samples, sections were contrasted with saturated solution of uranyl acetate in 50% ethanol for 30 min and in Reynold’s lead citrate for 5 min [115]. The stem and root sections were stained with 0.1% KMnO_4_ in citrate buffer for 45 min at room temperatures for lignin [116]. All sections were examined under Jeol JEM 2100 (Japan) TEM at 120 kV.

### 4.2. Ion Analysis

Ions such as Na^+^, K^+^, Ca^2+^ (cations) and Cl^−^ (anion) were analysed in root, stem, and leaves of SS (control), SSN (200 mM sodium-chloride-treated), ST (control), and STN (200 mM sodium-chloride-treated) plants. Plant samples were dried in a hot-air oven at 104 °C, powdered, and then each sample was digested with 8 mL of nitric acid along with peroxide. The suspension was filtered and diluted with 50 µL of deionized water. Concentrations of Na^+^, K^+^ and Ca^2+^ ions were determined using an inductively coupled plasma–atomic emission spectrometer. Cl^−^ was estimated via potentiometric titration of the tissue extract [117]. Units were expressed as mg/g dry weight of the tissue. Experiments were repeated two times. Each time, biological replicates were taken.

### 4.3. Plant Material and RNA Isolation

Seeds of *S. bicolor* variety ICSR-56 susceptible to salt stress (SS) and CSV-15 variety tolerant to salt stress (ST) were procured from ICRISAT, Patancheru, Hyderabad, India. Thirty-day-old ICSR-56 and CSV-15 sorghum seedlings were chosen for sodium chloride treatments. While control ICSR-56 and CSV-15 have been named as SS and ST, salt-susceptible seedlings treated with 200 mM sodium chloride solution for 24 h have been named SSN, and salt-tolerant seedlings treated with the same amount of sodium chloride have been named STN. After the completion of the treatment, all the four seedling samples (SS, SSN, ST, and STN) were collected and frozen immediately in liquid nitrogen and stored at −80 °C for total RNA extraction and gene expression analysis via quantitative real-time PCR (qRT-PCR). Two independent biological replicates for each tissue sample were harvested. Total RNA was isolated from the leaf samples using ZR Plant RNA Miniprep (ZYMO Research) as per the manufacturer’s instruction. RNA quality and quantity were checked on 1% denaturing RNA agarose gel and NanoDrop/Qubit fluorometer, respectively.

### 4.4. Nextseq Paired End (PE) Library Preparation and RNA-Seq

Transcriptomics is the study of the complete set of RNA transcripts that are produced by the genome, under specific circumstances or in specific cell-using high-throughput methods, such as microarray analysis. Comparison of transcriptomes allows the identification of genes that are differentially expressed in distinct cell populations or in response to different treatments. To augment this, the RNA-seq paired-end (PE) sequencing libraries were prepared using Illumina TruSeq Stranded mRNA sample Prep kit. From total RNA, mRNA was enriched using poly-T attached magnetic beads; first strand cDNA was synthesised using SuperScript II and Act-D mix. The first strand cDNA was then synthesized to the second strand using a second strand mix. The double-stranded cDNA was then purified using AMPure XP beads followed by A-tailing adapter and enriched via PCR. The PCR-enriched libraries were analyzed in the 4200 TapeStation system (Agilent Technologies) using high-sensitivity D1000 Screen tape. After quality and quantity check of paired-end libraries using Agilent 4200 TapeStation system, PE Illumina libraries were loaded onto NextSeq 500 for cluster generation and sequencing. Entire protocol (pipeline) is shown in the Appendix A. Paired-end sequencing allows the template fragments to be sequenced in both the forward and reverse directions on NextSeq. The kit reagents were then used in binding of samples to complementary adapter oligos on paired-end flow cells. The adapters were designed to allow selective cleavage of the forward strands after re-synthesis from the opposite end of the fragment.

Four libraries (SS, SSN, ST, and STN) were prepared from salt-treated leaf tissues of sorghum. Total RNA was isolated and processed for library preparation. The quality-control-passed RNA samples were then processed for library preparation (Appendix A). The paired-end (PE) libraries were prepared from total RNA by means of Illumina using TruSeq Stranded Standard mRNA sample Prep kit. The means of the library fragment size distributions are 394 bp, 373 bp, 387 bp, and 379 bp for SS, SSN, ST, and STN, respectively. The libraries were sequenced on NextSeq 500 using 2 × 75 bp chemistry. Sequenced raw data of 4 samples were processed to obtain high-quality clean reads using Trimmomatic v0.35 to remove adapter sequences, ambiguous reads (reads with unknown nucleotides “N” larger than 5%), and low-quality sequences (reads with more than 10% quality threshold (QV) <20 Pphred score). A minimum length of 50 nucleotides (nt) after trimming was applied. After removing the adapter and low-quality sequences from the raw data, high-quality reads were obtained (Appendix A). The high-quality (QV > 20) PE reads were used for reference-based read mapping.

### 4.5. Downstream Annotation and Bioinformatic Analysis

The reference genome of *S. bicolor*, with genome size of 709.3 Mb, and the gene feature format (GFF) file used for analysis were downloaded from NCBI. The reference fasta file and the GFF download links are as follows (ftp://ftp.ncbi.nlm.nih.gov/genomes/all/GCF/000/003/195/GCF (accessed on 13 March 2023) 000003195.3 *S. bicolor* NCBIv3/GCF 000003195.3 *S. bicolor* NCBIv3 genomic.fna.gz; ftp://ftp.ncbi.nlm.nih.gov/genomes/all/GCF/000/003/195/GCF (accessed on 13 March 2023) 000003195.3 *S. bicolor* NCBIv3/GCF 000003195.3 *S. bicolor* NCBIv3 genomic.gff.gz). For high-quality reads, raw reads were trimmed using the Trimmomatic tool [118]. The TopHat (https://ccb.jhu.edu/software/tophat/manual.shtml, accessed on 13 March 2023) and Cufflinks (version 2.2.1; http://cufflinks.cbcb.umd.edu/, accessed on 13 March 2023) software were used to align reads from an RNA-Seq experiment to a reference-based assembly [119,120] with default parameters (inner distance 150 and standard deviation of 100).

### 4.6. Downstream Annotation and Bioinformatic Analysis

The reference genome of *S. bicolor*, with genome size of 709.3 Mb, and the gene feature format (GFF) file used for analysis were downloaded from NCBI. The reference fasta file and the GFF download links are as follows (ftp://ftp.ncbi.nlm.nih.gov/genomes/all/GCF/000/003/195/GCF 000003195.3 (last accessed on 13 March 2023) [1] *S. bicolor* NCBIv3/GCF 000003195.3 *S. bicolor* NCBIv3 genomic.fna.gz; ftp://ftp.ncbi.nlm.nih.gov/genomes/all/GCF/000/003/195/GCF (last accessed on 13 March 2023) 000003195.3 *S. bicolor* NCBIv3/GCF 000003195.3 *S. bicolor* NCBIv3 genomic.gff.gz). For high-quality reads, raw reads were trimmed using the Trimmomatic tool [118]. The TopHat (https://ccb.jhu.edu/software/tophat/manual.shtml (last accessed on 13 March 2023)) and Cufflinks (version 2.2.1; http://cufflinks.cbcb.umd.edu/ (last accessed on 13 March 2023)) software were used to align reads from an RNA-Seq experiment to a reference-based assembly [119,120] with default parameters (inner distance 150 and standard deviation of 100).

### 4.7. DEG, KEGG and GO Enrichment and Pathway Analysis

Differential gene expression was determined using Cuffdiff utility provided in Cufflinks package using options, upper-quartile-norm, total-hits-norm, and frag-bias-correct. The transcripts with log2-fold change ≥ 1 (upregulated genes) and ≤(−1) (downregulated genes) with *p*-value cut off of ≤0.05 were considered as significantly differentially expressed transcripts.

### 4.8. GO Enrichment and Pathway Analysis

GO enrichment for various differentially expressed transcripts was performed using the BiNGO plug-in (version 3.0.3) at Cytoscape platform (version 3.2.1; http://www.cytoscape.org/, accessed on 13 March 2023). Rice GO information for biological process and molecular function was used for GO enrichment analysis. A *p*-value cut-off of ≤0.05 was considered significant, and we applied a hypergeometric test to identify enriched GO terms in BiNGO. Pathway analysis of differentially expressed transcripts was carried out using Mapman (version 3.5.1; http://mapman.gabipd.org/web/guest, accessed on 13 March 2023) with *p*-value cut-off of ≤0.05. The differentially expressed transcripts were mapped on Arabidopsis pathway genes to identify the transcripts involved in specific pathways.

### 4.9. Validation of DEGs by qRT-PCR Analysis

Twenty selected genes that are differentially expressed have been further validated using qRT-PCR on the ABI 7500 system (Applied Biosystems, MA, USA) for the genotypes SS and ST. The names of the genes and the forward and reverse primers used for qRT-PCR are given in the Appendix A. Total RNA was extracted from the seedlings treated with 200 mM sodium chloride using nucleo-spin plant RNA isolation kit (MACHERY-NAGEL) as per the instructions given. An amount of 3 µg of total RNA was translated into cDNA using a first-strand synthesis kit (Thermo Scientific, Waltham, MA, USA). SYBR Green Master Mix (2x, Takara, Shiga, Japan) was used as per the recommendations with the following thermal cycles: 1 cycle at 95 °C for 10 min, followed by 40 cycles alternatively at 95 °C for 15 s and 60 °C for a minute. After the 40th cycle, amplicon dissociation curves were recorded through raising the temperature from 58 to 95 °C within 20 min. Three biological replicates and two technical replicates were taken for analysis. Gene expression data were normalized with the sorghum genes Acyl Carrier Protein 2 (SbACP2) and Elongation Factor *p* (SbEF-*p*). Relative gene expressions were calculated through deploying REST software [121].

### 4.10. Statistical Analysis

All statistical analyses were carried out. For anatomical sections, two biological replicates and three technical replicates were taken. For ion analysis, average values from two independent experiments were taken. Each time, two biological replicates and three technical replicates were taken. The *p*-value heuristic (*p* < 0.05) was considered while inferring the DEGs and then sorted before they were validated using qRT-PCR. The mean and standard error (SE) from two independent experiments are shown. Asterisks * and ** indicate significant differences in comparison with SS and ST without sodium chloride treatment at *p* ˂ 0.05 and *p* ˂ 0.01, respectively.

## Figures and Tables

**Figure 1 plants-12-02400-f001:**
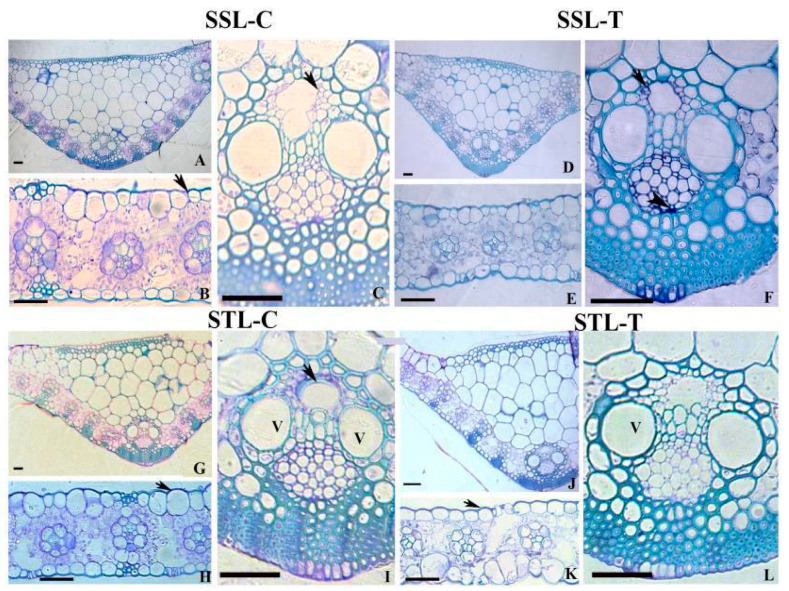
Transverse sections showing (**A**–**I**) anatomical characteristics in the leaf of control (SS and ST) and sodium-chloride-treated (SSN and STN) samples of sorghum plants. SS leaf tissue (SSL) (**A**–**E**) shows a relatively thin wall of epidermal cells (**B**) and thin and less lignified vessel elements (**C**) (arrow). Accumulation of polysaccharides was noticed in the cell corners of phloem elements within the vascular bundles. ST leaf (STL) (**G**–**L**) shows thicker cell wall in the epidermis (**H**) (arrow) and more lignified vessel walls (**I**). The increase in thickness of the upper cell wall of epidermis (**K**) (arrow) and thick-walled and highly lignified cell wall of vessel elements remain intact (**L**) in STN leaf. Scale bar = 50 µm. SS = salt-susceptible genotype ICSR-56; ST = salt-tolerant genotype CSV-15; SSN = salt-susceptible plants grown under sodium chloride stress; STN = salt-tolerant plants grown under sodium chloride stress; SSL-C = salt-susceptible leaf grown under control (without salt stress); SSL-T = salt-susceptible leaf grown under 200 mM sodium chloride stress conditions; STL-C = salt-tolerant leaf grown under control (without salt stress); STL-T = salt-tolerant leaf grown under 200 mM sodium chloride stress conditions; V = vacuole.

**Figure 2 plants-12-02400-f002:**
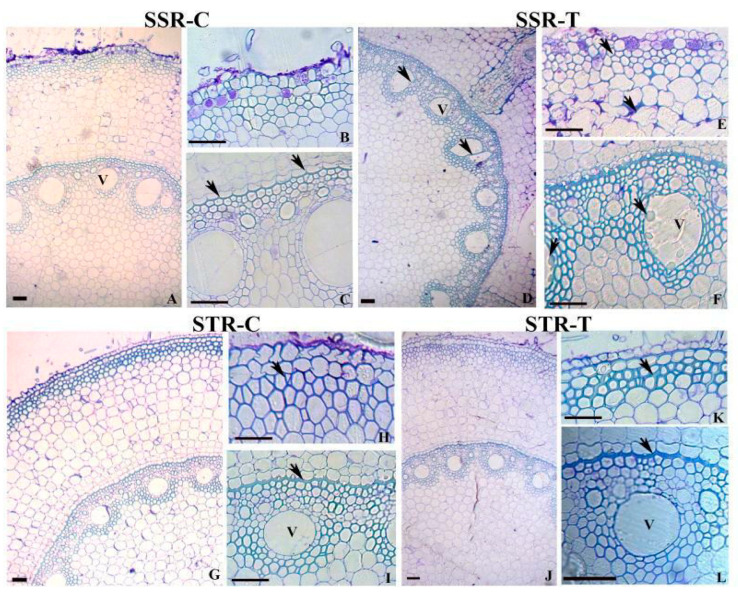
Transverse sections showing (**A**–**I**) anatomical characteristics in the root of SS, ST, and sodium-chloride-treated (SSN and STN) samples of sorghum plants. SS root tissue (SSR) (**A**–**E**) shows a thin lignified wall of outer cortical cells (**B**) and endodermis with thinner Casparian thickening (**C**) (arrows). Note the collapse of vessels (**D**), accumulation of polysaccharides in the cell corners of cortical cells (**E**), (arrow) and formation of tylosis in the vessels (**F**) (arrows) in the SSN roots. Salt-tolerant root (STR), (**G**–**L**) shows thick lignified walls of outer cortical cells (**G**) (arrow) and relatively thicker Casparian thickening of endodermis (**H**) (arrow). Note the enhanced lignification in the outer cortical cells (**J**), and Casparian thickening (**K**) (arrow) and intact vessel cell wall (**L**) in the STN roots. Scale bar = 50 µm. SS = salt-susceptible genotype ICSR-56; ST = salt-tolerant genotype CSV-15; SSN = salt-susceptible plants grown under 200 mM sodium chloride stress; STN = salt-tolerant plants grown under 200 mM sodium chloride stress; SSR-C = salt-susceptible root grown under control (without salt stress); SSR-T = salt-susceptible root grown under 200 mM sodium chloride stress conditions; STR-C = salt-tolerant root grown under control (without salt stress); STR-T = salt-tolerant leaf grown under 200 mM sodium chloride stress conditions; V = vacuole.

**Figure 3 plants-12-02400-f003:**
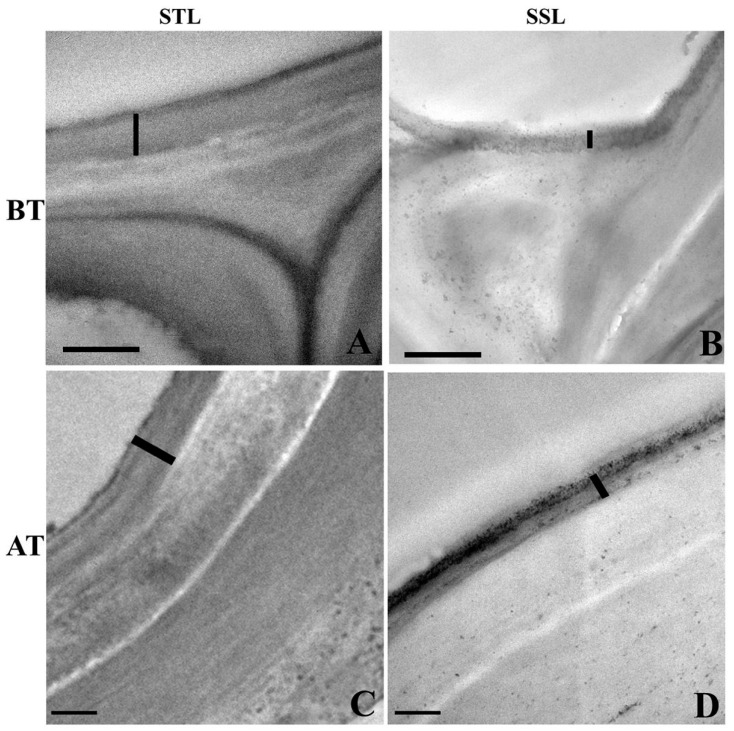
Transmission electron microscopy (TEM) images from the transverse sections of the upper epidermal cells from salt-tolerant leaf (STL) and salt-susceptible leaf (SSL) of *Sorghum bicolor* before treatment with sodium chloride (BT) and after treatment with sodium chloride (AT). (**A**) Salt-tolerant leaf showing a thick cuticle on the upper epidermal cell. (**B**) Relatively thinner cuticles are seen in the salt-susceptible leaf. (**C**) The cuticle of STL shows an increase in thickness of cuticle after salt treatment. (**D**) Thin cuticle in the SSL after salt treatment. Vertical bar indicates the cuticle thickness. Scale bar = 2 µm. SSL = salt-susceptible leaf; STL = salt-tolerant leaf.

**Figure 4 plants-12-02400-f004:**
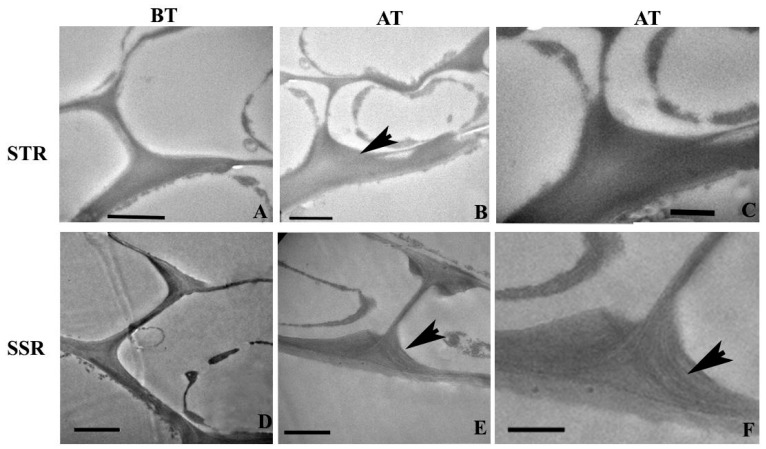
Transmission electron microscopy (TEM) images from the transverse sections from the endodermal cells in the root of salt-tolerant (STR) and salt-susceptible (SSR) plants of *Sorghum bicolor*. (**A**) Salt-tolerant root showing a relatively thicker Casparian strip. (**B**) Increase in the thickening of Casparian stipe (arrow) after salt treatment of STR. (**C**) Uniform electron density from Casparian thickening region of STR after KMnO_4_ contrasting due to homogenous lignin distribution. (**D**) Salt-susceptible root shows relatively thin Casparian thickening. (**E**) An abnormal pattern of thickening in the SSR after salt treatment (arrow). (**F**) The Casparian strip of SSR after salt treatment shows in homo genous lignin distribution as layers (arrow) within thickening. Scale bar = 2 µm. SSR = salt-susceptible root; STR = salt-tolerant root; BT = root sections before sodium chloride stress treatment; AT = root sections after sodium chloride stress treatment.

**Figure 5 plants-12-02400-f005:**
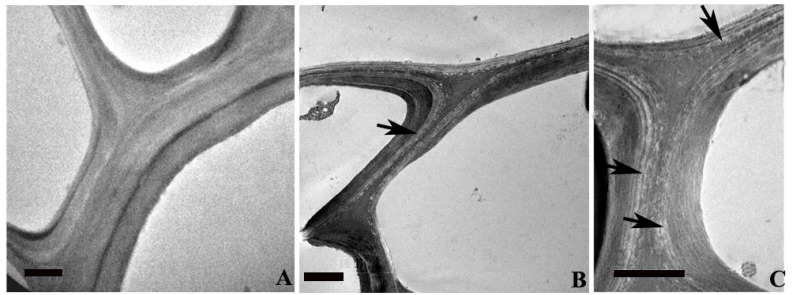
Transmission electron microscopy (TEM) images from the transverse sections from the vessel cell wall in the root tissue of salt-tolerant (**A**) and salt-susceptible (**B**,**C**) plant of *Sorghum bicolor*. (**A**) The thick cell wall of vessels from salt-tolerant root shows more or less homogenous lignin distribution. Note the high lignin distribution in the sub layer of S2 wall layer. (**B**) The thin wall of the vessel in the salt-susceptible root (SSR) shows less lignin distribution in the secondary wall layers. (**C**) Vessel wall of SSR is showing sub layers with inhomogeneous lignin distribution within S1 and S2 wall layers (arrows). Scale bar = 2 µm.

**Figure 6 plants-12-02400-f006:**
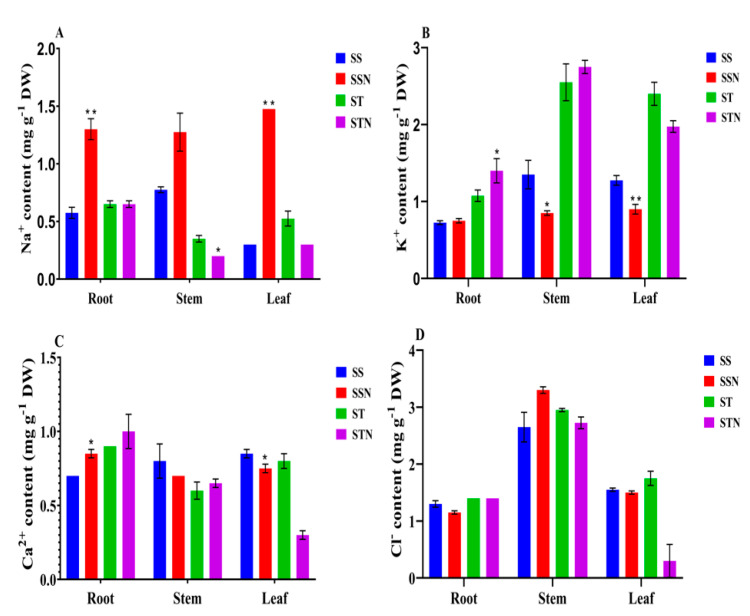
Ions such as sodium (Na^+^) (**A**), potassium (K^+^) (**B**), calcium (Ca^2+^) (**C**), and chloride (Cl^−^) (**D**) have been analysed in both salt-susceptible ICSR-56 (SS) and salt-tolerant CSV-15 (ST) sorghum genotypes in root, stem, and leaf tissues before salt stress (control) and after applying 200 mM sodium chloride stress for 24 h. While organs are represented on *X*-axis, ion values (mg/g DW) are represented on *Y*-axis. The mean and standard error (SE) from two independent experiments with three technical replicates are shown. * and ** indicate significant differences in comparison with SS and ST without sodium chloride treatment at *p* < 0.05 and *p* < 0.01, respectively.

**Figure 7 plants-12-02400-f007:**
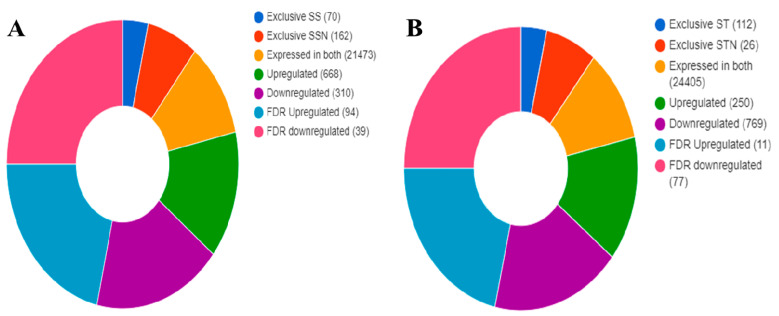
Differentially expressed genes (DEGs) associated with SS/SSN (**A**) and ST/STN (**B**) pairs were identified across the inferred datasets.

**Figure 8 plants-12-02400-f008:**
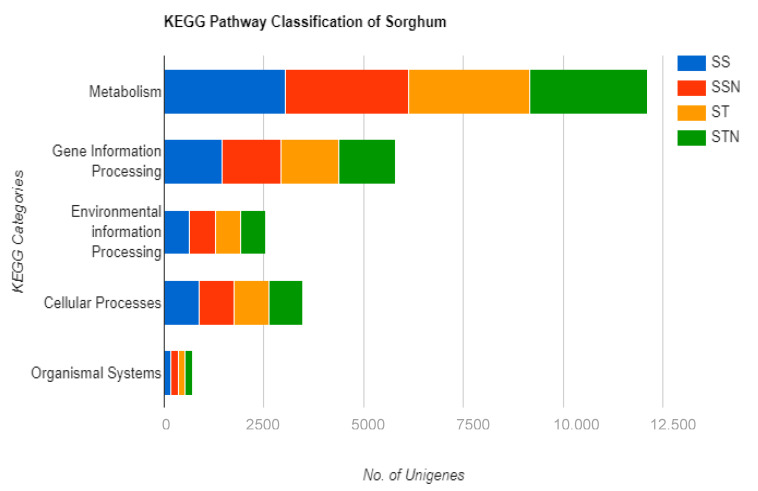
The KEGG pathway revealed diverse genes associated with carbohydrate, energy, lipid, nucleotide, amino acid, metabolism of cofactors and vitamins, terpenoids, polyketides, and secondary metabolism in SS, SSN, ST and STN samples. Genes implicated in gene information processing, signal transduction, cellular processes, and organismal systems have been detected.

**Figure 9 plants-12-02400-f009:**
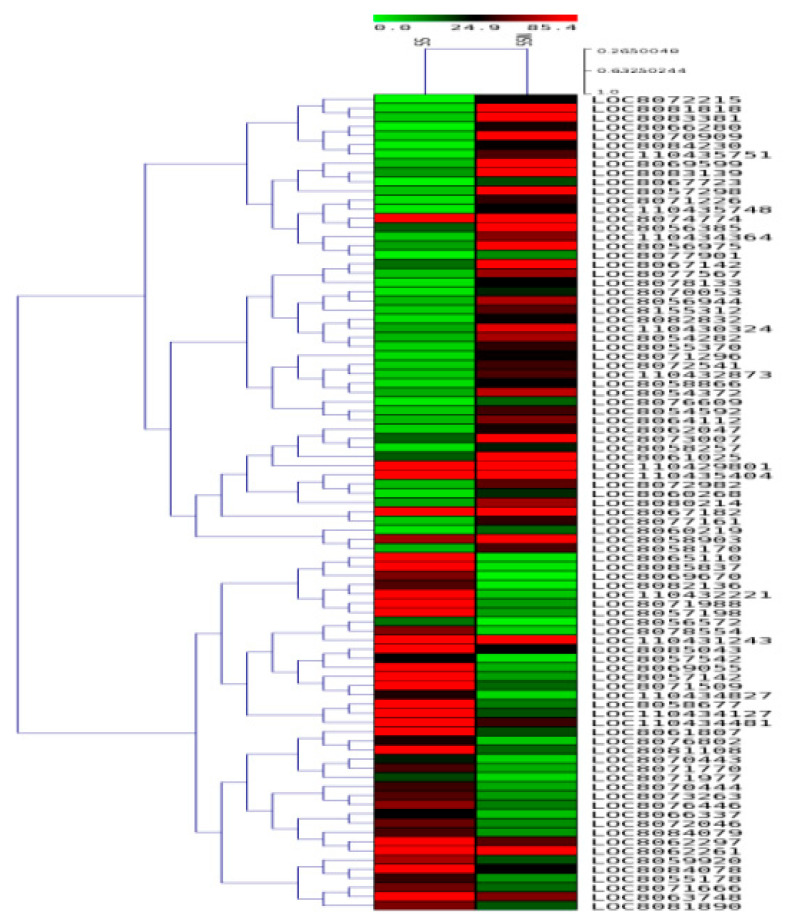
Heat map showing the differentially expressed genes in SS and SSN (salt-treated) sample combinations.

**Figure 10 plants-12-02400-f010:**
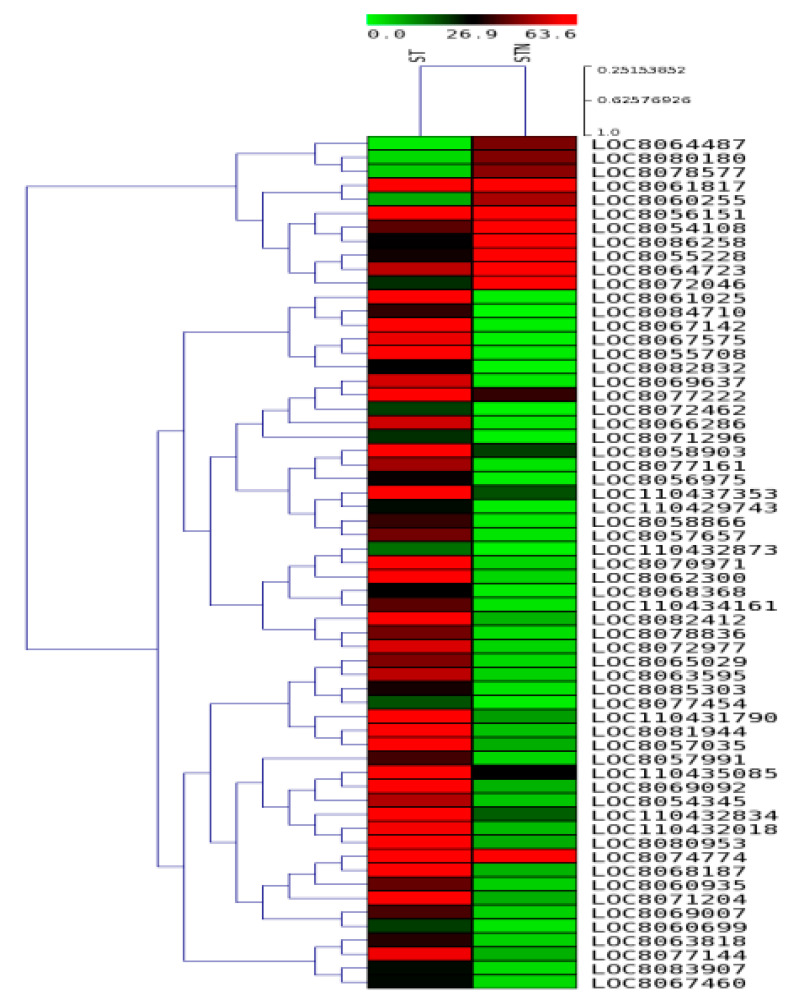
Heat map showing the differentially expressed genes in ST and STN sample combinations.

**Figure 11 plants-12-02400-f011:**
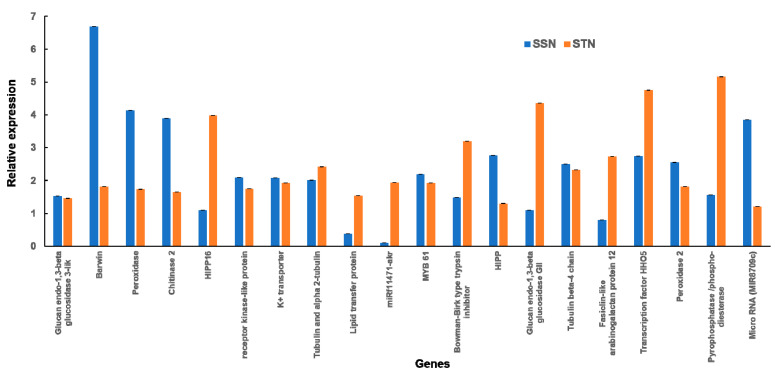
Expression analysis of the transcripts in salt-susceptible (SS) and salt-tolerant (ST) genotypes under 200 mM sodium chloride stress conditions. Values represent the expression fold obtained after normalizing against the reference genes. All samples were analysed in triplicate in three independent experiments. Different genes are represented on the *X*-axis, whereas relative expression is represented on the *Y*-axis. Standard error bars are shown.

**Table 1 plants-12-02400-t001:** GO analysis statistics between SS, SSN, ST, STN samples.

		Biological Process	Cellular Components	Molecular Function
**SS vs. SSN**	Downregulated	83	81	118
	Exclusive SS	6	6	7
	Exclusive SSN	29	36	35
	Expressed in both	6396	7340	7858
	False discovery rate (FDR) downregulated	14	13	16
	FDR upregulated	38	35	47
	Downregulated	239	245	296
**ST vs. STN**	Downregulated	299	321	343
	Exclusive ST	29	32	36
	Exclusive STN	2	2	2
	Expressed in both	6213	7136	7632
	FDR downregulated	32	32	39
	FDR upregulated	4	6	10
	Downregulated	89	96	120

**Table 2 plants-12-02400-t002:** Names of genes used for qRT-PCR analysis and their up- and downregulation in SS and ST under salt stress conditions.

S. No.	Name of Gene Selected for qRT-PCR	SSN/STN Up- and Down-Regulated/Function	References
1	Glucan endo-1,3-beta glucosidase 3-like Upregulated)	It stimulates and initiates the plant defence mechanism. A glucan endo-1,3-beta-glucosidase precursor has been detected in rice under salt stress. It plays a role during abiotic stress.	[37,38]
2	BARWIN (Upregulated)	BARWIN has similarity with proteins encoded by the wound-induced genes of barley (win) and others. It is involved in a common defence mechanism in plants.	[39]
3	Peroxidase 42 (Upregulated)	Peroxidases play prominent roles in antioxidant response through removing hydrogen peroxide from the system. Its overexpression results in salt stress tolerance.	[40]
4	Chitinase 2 (Upregulated)	Overexpression of a chitinase 2 gene (LcCHI2) in maize and tobacco improved salt and alkaline stress tolerance through reducing Na^+^ accumulation and malon-dialdehyde content.	[41]
5	Heavy metal-associated isoprenylated plant protein 16 (HIPP16) (Upregulated)	In addition to involvement in heavy metal homeostasis and detoxification, they are associated with abiotic stresses such as drought, cold, and plant–pathogen interactions.	[42]
6	Receptor kinase-like protein XA 21 (Upregulated)	XA 21 is an immune sensor protein, which promotes survival during dehydration stress. Its overexpression has enhanced lignin deposition and cellulose in the xylem vessels and surrounding cells.	[43]
7	K^+^ transporter (Upregulated)	Acquisition of K^+^ during salt stress is crucial. Its tissue-specific regulation of Na^+^ and K^+^ transporters explains genotypic differences in salt stress tolerance.	[44]
8	Tubulin and alpha 2-tubulin (Upregulated)	Microtubules and their dynamics play key roles in the adaptation of plants and their tolerance to salt stress.	[45]
9	Lipid transfer protein (LTP) (Downregulated)	LTPs are regulated by upstream transcription factors or kinase orphosphatases. They are critical for imparting salt/abiotic stress tolerance. However, it is downregulated in the SSN line but not in STN.	[46,47]
10	Micro RNA (miRf11471-akr) (Downregulated)	They are involved in gene regulation at the post-transcriptional level and associated with salt/abiotic stress tolerance. It is downregulated in SSN but not in STN.	[48]
11	MYB 61 (v-Myb avian myeloblastosis viral oncogene homolog transcription factor) (Upregulated)	Several MYBs have been shown to be associated with salinity stress in diverse plant systems.	[49,50,51,52,53]
12	Bowman-Birk type trypsin inhibitor (Upregulated)	It is involved in salt stress tolerance in wheat and other plants via limiting the transport of Na^+^ from the root to the shoot.	[54]
13	Heavy metal-associated isoprenylated plant protein (HIPP) (Upregulated)	Besides involvement in heavy metal homeostasis and detoxification, they are associated with abiotic stresses such as drought, cold, and plant–pathogen interactions.	[42]
14	Glucan endo-1,3-beta glucosidase (Upregulated)	It stimulates and initiates the plant defence mechanism. It plays a role in cell division, flower formation, and a vital role during abiotic stress.	[37,38]
15	Tubulin beta-4 chain (Upregulated)	Microtubules and their dynamics play key roles in the adaptation of plants and tolerance to salt stress.	[45]
16	Fasciclin-like arabinogalactan protein 12 (Downregulated)	This gene is involved in developing xylem and cell wall formation during salt stress. Additionally, it controls ROS levels. It is downregulated in STN, but not in SSN.	[11,55]
17	HHO5 (HRS1 homolog, (transcription factor) (Upregulated)	NIGT1/HRS1/HHO transcription factors are associated with salt stress tolerance, though the precise function is obscure.	[56]
18	Peroxidase 2 (Upregulated)	It removes ROS-like hydrogen peroxide generated during salt stress.	[57]
19	Pyrophospha-tase/phospho-diesterase (Upregulated)	It improves salt stress tolerance.	[58]
20	Micro RNA (ncRNA) (un-characterized) (miR8709c) (Upregulated)	It is suspected to be associated with *cis-trans* inter-conversion of cytokinin zeatin. Its overexpression led to decreased FAD and enhanced FMN and riboflavin contents, but its precise function is not known.	[48]

## Data Availability

All the data associated with the transcriptome sequence are available via Bioproject: PRJNA930967.

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
