# Peer review of "Root and Leaf Anatomy, Ion Accumulation, and Transcriptome Pattern under Salt Stress Conditions in Contrasting Genotypes of Sorghum bicolor"

_plants, 2023, doi:10.3390/plants12132400_

Round 1

Reviewer 1 Report

Article Transcriptome, root and leaf anatomy and ion accumulation pattern under salt stress conditions in contrasting genotypes of Sorghum bicolor by Appa Rao Karumanchi, Pramod Sivan, Divya K, G Rajasheker, S Anil Kumar, Sudhakar Reddy P, Prashanth Suravajhala , Sudhakar Podha, Kavi Kishor P B is a comprehensive study in which the anatomical and expression analysis of two sorghum genotypes differing in resistance was carried out.

The work is somewhat chaotic.

There are significant claims to the materials and methods and to the presentation of the results.

So, images must have a signature from which everything is clear: the method, what is compared with what, what the signatures mean. However, in Figures 1 and 2, the thickness of the sections is not indicated, the method of contrasting is not described, but the most interesting thing is that in the photographs, by the way, of good quality with high-quality sections, there are letters that are not reflected in the signatures, for example V. No less strange is the use of undeciphered abbreviations SS and ST. Why the rulers of different thicknesses is impossible to understand.

There was also an incident with electron microscopic photographs in Figures 3,4,5, as the signatures do not indicate the type of tissue to which the cells under study belong, although of course you can guess, but not everyone can and should guess ... It’s better to sign anyway, and it’s especially important to indicate the epidermis of which side of the leaf is shown and what these cells are called ... The ruler in Figure 3D is not perpendicular ...

Apparently, the confusion with the tissues and structure of plants persists in the description of the method for isolating NA. Here the authors completely forget that the leaf is an organ, not a tissue. Among other things, it is not possible to understand how many and at least how many technical repetitions were in this study.

I think the authors should carefully look at the work and correct errors in the design.

I recommend writing NaCl in full in signatures, drawings and diagrams, as it is short, but understandable without explanation.

You just need to carefully look at the manuscript and drawings.

Author Response

The work is somewhat chaotic.

There are significant claims to the materials and methods and to the presentation of the results.

 Ans: Material and methods and also the results have been partly revised as per the suggestion.

So, images must have a signature from which everything is clear: the method, what is compared with what, what the signatures mean. However, in Figures 1 and 2, the thickness of the sections is not indicated, the method of contrasting is not described, but the most interesting thing is that in the photographs, by the way, of good quality with high-quality sections, there are letters that are not reflected in the signatures, for example V. No less strange is the use of undeciphered abbreviations SS and ST. Why the rulers of different thicknesses is impossible to understand.

 Ans: The figures and the figure legends have been corrected as per the suggestions. The letters in the figures have been corrected as ‘upper case’ in all figures. Scale bar is indicated for all figures. Full names for all abbreviations have been given in foot notes of Figures 1, and 2 as suggested.

SS = salt susceptible genotype; ST = salt tolerant genotype; SSN = salt susceptible plants grown sodium chloride stress; STN = salt tolerant plants grown under sodium chloride stress; SSL-C = salt susceptible leaf grown under control (without salt stress); SSL-T = salt susceptible leaf grown under sodium chloride stress conditions; STL-C = salt tolerant leaf grown under control (without salt stress); STL-T = salt tolerant leaf grown under sodium chloride stress conditions; SSR-C = salt susceptible root grown under control (without salt stress); SSR-T = salt susceptible root grown under sodium chloride stress conditions; STR-C = salt tolerant root grown under control (without salt stress); STR-T = salt tolerant leaf grown under sodium chloride stress conditions; V = vacuole.

There was also an incident with electron microscopic photographs in Figures 3, 4, 5, as the signatures do not indicate the type of tissue to which the cells under study belong, although of course you can guess, but not everyone can and should guess ...

 Ans: Figure 3 represents salt susceptible leaf (SSL) and salt tolerant leaf (STL) sections.

Figure 4 represents salt susceptible root (SSR) and salt tolerant root (STR) sections.

Figure 5 represents transverse electron micrographs from the transverse sections of root of salt tolerant (A) and salt susceptible (B, C) genotypes of S. bicolor. SS = salt susceptible genotype; ST = salt tolerant genotype; SSN = salt susceptible genotype after sodium chloride stress; STN = salt tolerant genotype after sodium chloride stress; SSR = salt susceptible root, STR = salt tolerant root, SSL = salt susceptible leaf, STL = salt tolerant leaf; BT = leaf or root sections taken before sodium chloride treatment; AT = leaf or root sections taken after sodium chloride treatment. This information is shown in the legends for figures.

It’s better to sign anyway, and it’s especially important to indicate the epidermis of which side of the leaf is shown and what these cells are called ...

 Ans: The cuticle from the upper epidermal cells is represented in Figure 3, endodermal cells in Figure 4 and vessels cell wall in Figure 5. The information on the type of cells has been added in the legend. All abbreviations have been expanded.

The ruler in Figure 3D is not perpendicular ...The orientation of ruler has made perpendicular with respect to the tissue orientation in the electron micrograph.

 Ans: The orientation of ruler has been made perpendicular with respect to the tissue orientation in the electron micrograph.

Apparently, the confusion with the tissues and structure of plants persists in the description of the method for isolating NA. Here the authors completely forget that the leaf is an organ, not a tissue.

 Ans: Yes, we agree that leaves and roots are organs. Every organ is made up of tissues and in histological preparations they are termed as ‘leaf tissue’, ‘root tissue’ etc. Therefore, for anatomical section cuttings, general terms like leaf sections and root sections were used. For DNA isolation also, we used the terms “leaf” and “root” instead of tissues.

Among other things, it is not possible to understand how many and at least how many technical repetitions were in this study.

 Ans: For anatomical sections, ion analysis, and qRT-PCR analysis, two biological replicates and three technical replicates were used. For transcript analysis, two biological replicates were taken for salt susceptible (SS) and salt tolerant (ST) sorghum genotypes.

I think the authors should carefully look at the work and correct errors in the design.

 Ans: We have revised the manuscript carefully keeping in view of the reviewer’s suggestion. Errors if any have been corrected.

I recommend writing NaCl in full in signatures, drawings and diagrams, as it is short, but understandable without explanation.

 Ans: NaCl has been replaced with sodium chloride in the text, and in legends for figures also.

You just need to carefully look at the manuscript and drawings.

 Ans: We have gone through the manuscripts, made necessary changes and figures also have been modified as per the suggestions.

Reviewer 2 Report

The manuscript by Karumanchi et al. is trying to reveal the intricacy of salt response/tolerance in two contrasting sorghum varieties by studying their differences/alterations in cell wall anatomy, ion accumulation and transcriptomic abundance. It is a good idea to use a systematic approach with an attempt to reveal the basis salt stress tolerance in sorghum. However, this study suffers from a lack of scientific soundness and its experiments are not well designed or implemented. In addition, the manuscript needs substantial restructuring/revision to improve its clarity and readability.

Since the review file doesn’t provide line numbers, I listed a few following places in the abstract and introduction as examples.

“Conversely, roots with thick, lignified cell walls in hypodermis and endodermis were noticed in salt tolerant (ST) plants which further increased with the addition of NaCl (STN). Lignin distribution in the secondary cell-wall of sclerenchymatous cells beneath the lower epidermis was higher in ST leaf compared to SS genotype.” These sentences are very confusing. What was increased? Root mass? Cell wall thickness? Lignification level? Similarly, what was higher? Lignin content? 

To identify salt-regulated changes in gene expressions that are unique and genotype-dependent, transcriptomic analysis of the SS and ST sorghum genotypes exposed to 24h NaCl (200 mM) stress was carried out. Transcriptomic analysis unravelled 70 and 162 differentially expressed genes (DEGs) exclusive to SS and SSN. Sentences like this can be written more concisely to present the key findings.

The sentences in the third paragraph lack cohesion, with abrupt shifts in topic or facts. Please revise and restructure the sentences like those throughout the manuscript to enhance the logic flow and readability.

I have a major concern of the transcriptomic study and RT-PCR verification.

 First, the genotypic background of those two sorghum varieties was not mentioned anywhere in the manuscript. Names like ICSR-56 and CSV-15 basically tell our readers nothing. Their inherent differences should be first compared if any stress experiment of RNA-seq was conducted. How did they differ at the transcriptomic level before subjecting to salt stress treatment? Do they have similar development/growth at sampling? Please compare the DEGs between SS/ST before interpreting Figure 7-10. In addition, all those technical results (Table 1-3 et.al) should be moved to supplementary.

Secondly, it is very odd to present qRT-PCR data in a heat-map format (Figure 11). They should be presented with relative abundance with significance analysis results for the purpose of verification.  

Lack of logical flow; lack of clear and concise writing.

Author Response

The manuscript by Karumanchi et al. is trying to reveal the intricacy of salt response/tolerance in two contrasting sorghum varieties by studying their differences/alterations in cell wall anatomy, ion accumulation and transcriptomic abundance. It is a good idea to use a systematic approach with an attempt to reveal the basis salt stress tolerance in sorghum.

However, this study suffers from a lack of scientific soundness and its experiments are not well designed or implemented. In addition, the manuscript needs substantial restructuring/revision to improve its clarity and readability.

 Ans: The manuscript has been revised substantially and restructured and improved the readability.

Since the review file doesn’t provide line numbers, I listed a few following places in the abstract and introduction as examples.

“Conversely, roots with thick, lignified cell walls in hypodermis and endodermis were noticed in salt tolerant (ST) plants which further increased with the addition of NaCl (STN). Lignin distribution in the secondary cell-wall of sclerenchymatous cells beneath the lower epidermis was higher in ST leaf compared to SS genotype.” These sentences are very confusing. What was increased? Root mass? Cell wall thickness? Lignification level? Similarly, what was higher? Lignin content? 

 Ans: The secondary wall thickness and the number of lignified cells in the root hypodermis increased after the sodium chloride treatment.

To identify salt-regulated changes in gene expressions that are unique and genotype-dependent, transcriptomic analysis of the SS and ST sorghum genotypes exposed to 24h NaCl (200 mM) stress was carried out. Transcriptomic analysis unravelled 70 and 162 differentially expressed genes (DEGs) exclusive to SS and SSN.”  Sentences like this can be written more concisely to present the key findings.

 Ans: Thank you. The sentences have been modified in the abstract. In the salt susceptible (SS) genotype, 70 and 162 DEGs have been found expressed in the control (without sodium chloride treatment) and under sodium chloride stress respectively. In the salt tolerant (ST) genotype, 112 and 26 DEGs have been found expressed in the control (without sodium chloride treatment) and under sodium chloride stress respectively.

The sentences in the third paragraph lack cohesion, with abrupt shifts in topic or facts. Please revise and restructure the sentences like those throughout the manuscript to enhance the logic flow and readability.

 Ans: The sentences in the third paragraph of Introduction have been rewritten with more cohesion and with improved readability. Introduction has been revised to enhance the flow.

I have a major concern of the transcriptomic study and RT-PCR verification.

First, the genotypic background of those two sorghum varieties was not mentioned anywhere in the manuscript. Names like ICSR-56 and CSV-15 basically tell our readers nothing.

 Ans: Seeds of S. bicolor genotype ICSR-56, a relatively salt susceptible genotype, abbreviated as SS and the genotype CSV-15 as highly salt tolerant, abbreviated as ST were procured from ICRISAT, Patancheru, Hyderabad, India. This information has been furnished in Material and Methods. Thirty-day-old ICSR-56 and CSV-15 sorghum seedlings were chosen for sodium chloride treatments. Seedlings of these two were treated with 200 mM sodium chloride solution for 24-hours and have been named as SSN (salt susceptible treated with 200 mM sodium chloride) and STN (salt tolerant treated with 200 mM sodium chloride). This has been mentioned in Material and Methods section of the original manuscript.

Their inherent differences should be first compared if any stress experiment of RNA-seq was conducted. How did they differ at the transcriptomic level before subjecting to salt stress treatment?

 Ans:  ICSR-56 and CSV-15 have been named as relatively salt susceptible (SS) and highly salt tolerant (ST) sorghum genotypes respectively. We obtained the seeds from the reliable source of ICRISAT, Hyderabad. We performed transcriptomic analysis of these two genotypes, and found distinct differentially expressed genes (DEGs). Our objective was to see the differences in SS and SSN, ST and STN. Likewise, 668 and 250 have been found upregulated, 310 and 769 down-regulated in SS and ST genotypes. In salt susceptible untreated genotype and salt tolerant genotype (without sodium chloride stress), 70 and 112 DEGs were recorded respectively. This information is given in the results section of the revised manuscript.

Do they have similar development/growth at sampling? Please compare the DEGs between SS/ST before interpreting Figure 7-10. In addition, all those technical results (Table 1-3 et.al) should be moved to supplementary.

 Ans: Both salt susceptible (ICSR-56) and salt tolerant (CSV-15) genotypes had similar development/growth rates at the time of sampling. The purpose of this study is to compare between SS versus SSN and ST versus STN and not SS versus ST. Table 1 has been taken as a supplementary Table. Tables 2 and 3 remained in the revised version of the manuscript.

Secondly, it is very odd to present qRT-PCR data in a heat-map format (Figure 11). They should be presented with relative abundance with significance analysis results for the purpose of verification.  

 Ans: As suggested by the reviewer, we have represented the qRT-PCR data in the form of bar diagrams in the revised version of the manuscript.

Differential expression profile of the genes (Table 2) has been observed in SS and ST under 200 mM sodium chloride treatment. All the genes (Table 2) displayed higher expression levels in ST under sodium chloride treatment (Figure 11). Similarly, all the genes (Table 2) displayed higher expression levels in SS under sodium chloride treatment except lipid transfer protein (LTP), non-coding RNA (miRf11471-akr), and fasiclin-like arabinogalactan protein 12 are down-regulated (Figure 11).

Author Response

All abbreviations should be first identified before use them even if they were in abstract or another part of the manuscript

Ans: Necessary changes have been made as suggested by the reviewer. Material and Methods contain the full forms of all abbreviations.

There is an excessive use of acronyms, which makes the reading sometimes difficult

Ans: The acronyms like SS, SSN, ST, and STN were given the full form when used for the first time, and subsequently acronyms were used. The NaCl has been replaced with sodium chloride in the revised version as suggested.

The Abstract is general and not informative. The Abstract needs modification. Moreover, the authors should focus on what was the problem, the hypothesis, the treatments, main results and conclusion. All abbreviations should be first identified before use them even if they were in abstract or another part of the manuscript

Ans: It is not known if anatomical changes exist in the roots and leaves of salt susceptible and salt tolerant lines of Sorghum bicolor. Further, ion accumulation patterns and transcriptomic changes if any have not been thoroughly investigated between the salt susceptible and tolerant lines of S. bicolor. Accordingly, the leaves and roots of salt susceptible (SS), salt susceptible plants treated with sodium chloride (SSN), salt tolerant (ST), and salt tolerant genotype treated with sodium chloride (STN) have been used for finding out the differences in the root and leaf anatomy, ion [sodium (Na+), potassium (K+), magnesium (Mg2+), chloride (Cl-)] accumulation pattern in different organs like root, stem and leaf and differentially accumulated genes among the two contrasting genotypes under control and sodium chloride treated conditions. All the suggested changes have been made in the revised manuscript.

The introduction does not point out the gap of the literature the study seeks to fill and novelty of the study over the existing literature. This point showed be further elaborated. 

Ans: Information on the physiology of salt stress tolerance in S. bicolor is known (Nawaz et al. 2010, Bavi et al. 2011). Transcriptomic changes in the drought tolerant cultivars and also grain sorghum and sweet sorghum are available. But, it is not known if anatomical changes exist in the roots and leaves of the salt susceptible and salt tolerant lines of Sorghum bicolor. Further, cation and anion accumulation patterns in different organs and transcriptomic changes if any have not been thoroughly investigated between the salt susceptible and salt tolerant lines of S. bicolor. Accordingly, the leaves and roots of salt susceptible (SS), salt susceptible treated with sodium chloride (SSN), salt tolerant (ST), and salt tolerant genotype treated with sodium chloride (STN) have been used for finding out the differences in the leaf and root anatomy, ion accumulation pattern and differentially accumulated genes among the two lines. Transcriptomic profiling in relation to heat and drought stress was studied in grain sorghum (Johnson et al. 2014) and the same in two sweet sorghum genotypes with different salt tolerance abilities (Sui et al. 2015, Chen et al. 2022). But little is known about the anatomical changes, ion (cation and anion) accumulation patterns in different organs and transcriptomic differences that occur during short-term exposure to salt stress in S. bicolor genotypes with different salt tolerance abilities. Therefore, in the present study, we examined the changes in gene expressions during short-term salt stress exposure of seedlings using RNA sequencing (RNA-seq) and found out the candidate genes and pathways. Structural differences in the root and leaf anatomy and significant differences in ion accumulation patterns in different organs of the contrasting genotypes of sorghum under salt stress conditions have been unravelled. Such traits might help the breeders to generate cultivars that can withstand salt stress conditions.

The objectives and conclusion of the study are not clear and need to re-write

 Ans: Objectives: The objective of the study was to find out the leaf and root anatomical differences between contrasting genotypes of S. bicolor that differ in their tolerance levels to sodium chloride stress. Also, to know the ion (sodium, potassium, chloride and magnesium) accumulation patterns in different organs and the transcriptomic differences if any between the genotypes that differ in salt stress tolerance abilities of grain sorghum.

 Conclusion: It is concluded that significant changes in the leaf and root anatomy occur besides ion accumulation patterns and transcriptomic changes in the contrasting genotypes of grain sorgum. Such traits can used in the breeding programs aimed at developing the salt tolerant genotypes.

A relevant hypothesis for the study is missing from the introduction. A true scientific question should be formed

 Ans: Physiological changes in response to salt stress occur in Sorghum bicolor. But what is elusive so far is if anatomical changes occur in the roots and leaves of contrasting genotypes that differ in their salt stress tolerance abilities. Do anatomical changes contribute to the salt stress tolerance? Do contrasting genotypes differ in their ion accumulation patterns in the organs and in their differentially expressed genes in the grain sorghum lines? We tried to answer these questions in S. bicolor.

Simplify the statement in the paper. Please combine and condense the discussion and conclusion

Ans: The statement has been simplified. The discussion has been condensed. Conclusions have been combined with the discussion.

Round 2

Reviewer 1 Report

The manuscript, after serious revision, began to look much better. This article may be accepted for publication in this form.

Reviewer 2 Report

Thanks to the authors for addressing some of my comments. However, they didn't provide clear responses to my major concerns. The overall quality of data presentation is limited. For example, as a key finding of this study, Figure 11 is presented without any statistical analysis. It is very hard for our readers to accept their claims and conclusions, such as "qRT-PCR was performed to validate 20 DEGs in each SSN and 41 STN samples which confirm the transcriptomic results." Another example is that the background information for those two sorghum genotypes which I requested in my first round of review is still missing. Thus, even the "constracting genotypes" in the manuscript title is an unsupported statement. 

A cohensive, clear writing is still needed. Many sentences, for example, line 106 "Information on the physiology of salt stress tolerance in S. bicolor is known" sounds too vague and thus should be rewritten to the point that is closely related to the topic of this study.

Reviewer 3 Report

Dear Editor Journal of Plants MDPI

Manuscript ID: plants-2394184

I re-reviewed the manuscript Transcriptome, root and leaf anatomy and ion accumulation pattern under salt stress conditions in contrasting genotypes of Sorghum bicolor" again and the authors made all the amendments that I asked before so I think the manuscript is suitable for publishing

Regards